# Improving Developer Emotion Classification via LLM-Based Augmentation

## Abstract

Detecting developer emotion in the informative data stream of technical commit messages is a critical task for gauging signals of burnout or bug introduction, yet it exposes a significant failure point for large language models whose emotion taxonomies are ill-suited for technical contexts in the field of software engineering. To address this, the study introduces a dataset of 2,000 GitHub commit messages that have been human-labeled with a four-label scheme tailored for this domain: **Satisfaction, Frustration, Caution,** and **Neutral**. A diagnostic zero-shot evaluation of five pretrained models yields near-chance Macro-F1 (0.13–0.21) and systematic biases. While fine-tuning a code-aware encoder (CodeBERT) establishes a strong baseline (Macro-F1 $\approx 0.59$), this study introduces **CommiTune**, a simple hybrid method that first fine-tunes a LLaMA model on the manually labeled dataset, uses it to generate augmented data, and then fine-tunes CodeBERT on this expanded set, achieving Macro-F1 $\approx 0.82$ (Accuracy $\approx 0.81$) on an untouched test split. This demonstrates that hybrid augmentation can effectively repair the representation gap in technical emotion detection. These results establish reproducible training and validation schemes for software-engineering NLP. The code, prompts, and label mappings will be released upon acceptance.

## 1 Introduction

Software development is an intensely collaborative process, and commit messages form the central record of how developers describe and rationalize changes to code. Beyond documenting functionality, these short messages often encode subtle signals of developer emotion—from satisfaction when a feature is completed to frustration when a bug resists resolution. Detecting such emotions has practical value: it can serve as an early-warning system for burnout Sinha et al. (2016), highlight points of friction that introduce defects Yadav & Vishwakarma (2020), and ultimately improve developer productivity and project stability.

Despite this importance, emotion detection in commit messages exposes a fundamental limitation of current language models. Mainstream emotion taxonomies and datasets, often sourced from social media or a mix of informal, non-professional repositories, fail to capture the signals present in the terse, technical registers of high-quality software development Guo (2022). As a result, large pretrained models underperform Majumder et al. (2019). Our diagnostic study finds that zero-shot emotion classification on commit data yields near-chance Macro-F1 (0.13–0.21), with systematic biases such as over-predicting **Caution** from technical jargon and **Satisfaction** from verbs like "fix" or "implement." This gap highlights the need for domain-adapted representations that integrate both code-awareness and affective sensitivity.

To address this challenge, this work makes three contributions. First, a dataset of 2,000 human-labeled GitHub commit messages is introduced, annotated under a domain-appropriate four-label scheme: **Satisfaction, Frustration, Caution,** and **Neutral**. A transparent reconciliation pipeline prioritizes risk signals, improving inter-annotator agreement from $\kappa = 0.596$ to $\kappa = 0.710$. Second, baseline experiments show that fine-tuning a code-aware encoder (CodeBERT)(Feng et al., 2020) achieves Macro-F1 $\approx 0.59$, outperforming language-only baselines but leaving a representation gap. Finally, we present **CommiTune**, a simple hybrid pipeline that fine-tunes a LLaMA model Touvron et al. (2023) on the labeled dataset, uses it to generate augmented training examples, and then fine-tunes CodeBERT on this expanded corpus. CommiTune substantially improves performance, achieving

Macro-F1 $\approx 0.82$ (Accuracy $\approx 0.81$) on a held-out test set, with the largest gains in detecting **Satisfaction**.

These results demonstrate that hybrid augmentation can repair the failure modes of large language models in technical emotion detection. Beyond benchmarks, this work establishes reproducible protocols for annotation, reconciliation, and augmentation in software-engineering NLP. We highlight the broader lesson that adapting affective modeling to technical domains requires bridging both linguistic and representational gaps.

## 2 LITERATURE REVIEW

### 2.1 LLMS AND SENTIMENT/EMOTION ANALYSIS

Large Language Models (LLMs) such as BERT, GPT-3/4, and Flan-UL2 achieve strong results in sentiment and emotion recognition due to their contextual depth and transferability. While they generalize well in zero/few-shot settings, domain-specific fine-tuning often surpasses general-purpose modelsAcheampong et al. (2021). Earlier approaches with CNNs and LSTMs also effectively captured context in short texts, achieving over 90% accuracy in benchmark datasetsSinha et al. (2016)Majumder et al. (2019)Huq et al. (2020). BERT-based models surpass lexicon methods, but challenges remain in sarcasm detection, short texts, and imbalanced datasets (Guo, 2022). Moreover, pre-trained models can encode biases in prompt design, category selection, and training corpora. They perform well in binary sentiment classification but struggle with fine-grained or culturally dependent emotional distinctions Wankhade et al. (2022).

### 2.2 SENTIMENT ANALYSIS IN SOFTWARE ENGINEERING AND SOCIAL MEDIA

Building on these advances in general NLP, research has explored sentiment analysis in software engineering. Analyses of GitHub commits reveal that developer emotions are correlated with productivity, bug introduction, and burnout. Large-scale studies show most commit messages are neutral, but negative emotions peak in bug-related commits (Nandwani & Verma, 2021)Babu & Kanaga (2022). While LLMs offer strong baselines, smaller fine-tuned models (e.g., RoBERTa) often outperform in domain-specific tasks (Zhang et al., 2023). Parallel findings in social media research reinforce these patterns: machine learning models (e.g., GLM with 92% accuracy) capture affect effectively but struggle with informal text, sarcasm, and multimodal cues (Yadav & Vishwakarma, 2020)Majumder et al. (2019)Yang et al. (2024). These limitations echo the difficulties encountered in analyzing technical commit messages.

### 2.3 METHODS AND CHALLENGES

Across both general NLP and domain-specific studies, sentiment analysis methods span lexicon-based, machine learning, and hybrid approaches. While ML and transfer learning outperform static lexicons, challenges persist in ambiguity, sarcasm, multilingual settings, and limited domain datasets (Feldman, 2013)Acheampong et al. (2021)Majumder et al. (2019). This indicates that despite progress, robust emotion detection in technical contexts still requires methods tailored to the domain.

### 2.4 DATA AUGMENTATION WITH GENERATIVE MODELS

A critical challenge in sentiment analysis of commit messages is class imbalance, i.e., most commits are neutral, with relatively few expressing frustration or caution. Data augmentation has emerged as a solution. Recent studies explore generative models for augmenting scarce emotion data. Back-translation and paraphrasing with transformer models (Fadaee et al., 2017) increase diversity, while more recent approaches leverage GPT-style models to synthesize realistic training samples. Conditional text generation enables creation of domain-specific emotional data (Wang et al., 2023), improving balance across categories. In software engineering, synthetic commits generated with contextual cues could mitigate underrepresentation of rare emotions like "satisfaction" or "caution." Generative augmentation is not without risks: synthetic data may introduce noise or amplify biases. Nonetheless, careful prompt engineering and filtering strategies have shown improvements in

downstream emotion detection tasks (Wang et al., 2023). Incorporating augmentation with LLMs represents a promising direction for addressing imbalance in commit message sentiment analysis.

## 3 DATASET AND ANNOTATION

The foundation of this research is a dataset curated through a multi-stage process designed to ensure statistical validity and high-quality annotations. This process encompassed data sourcing and stratified sampling, a comprehensive text preprocessing pipeline, and an iterative annotation methodology to construct a reliable gold standard dataset.

### 3.1 DATA SOURCING AND SAMPLING STRATEGY

The source data for this study is a large-scale public dataset from Kaggle, comprising 4.3 million commit messages. These messages were extracted from 34 of the most popular and actively maintained projects on GitHub, selected based on their star and fork counts. This selection criterion ensures that the dataset is representative of high-quality, professional software development practices rather than casual or hobbyist projects. The dataset's utility for research is enhanced by its Open Data Commons Attribution License (ODC-By) v1.0, which guarantees transparency and allows for unrestricted use and reproduction of studies.

Given the computational expense of analyzing the entire 4.3 million commits, a representative subset of 20,000 messages was created. To avoid sampling bias and ensure the subset accurately reflected the parent corpus's diversity, a stratified sampling strategy was employed. This method involved creating a composite stratification label for each commit based on three features: the repository name, the project's overall size (defined by its total commit count), and a temporal grouping that assigned each commit to a specific calendar quarter (e.g., "2020Q1"). For instance, a commit with the label "100_python_2020Q1" would belong to the Python repository (with a total project size of 100,000 commits) and was made in the first quarter of 2020. By calculating the proportional representation of each stratum in the full dataset, we could then draw a random sample from each group, resulting in a 20,000-commit subset that preserved the original distribution of commits across projects of varying scales and time periods.

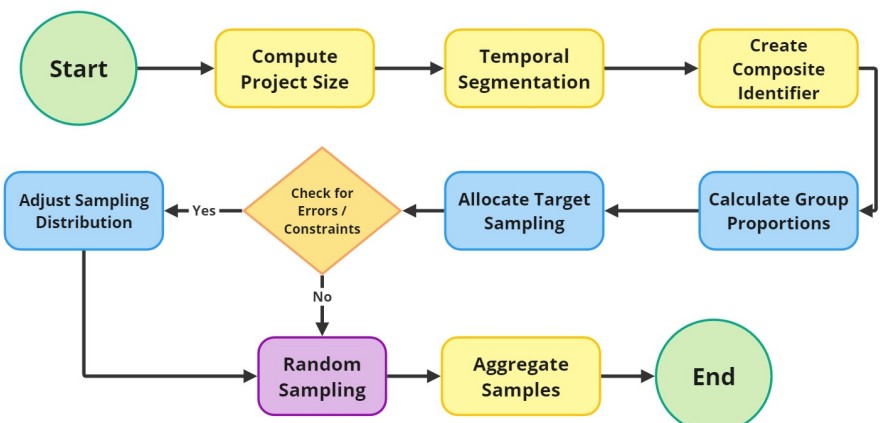

Figure 1: Stratified sampling pipeline for constructing the 20k subset from the 4.3M commits.

### 3.2 PREPROCESSING PIPELINE

Prior to annotation and analysis, the 20,000 sampled commits were subjected to a rigorous, multi-stage preprocessing pipeline to clean and standardize the raw text. The objective was to eliminate noise and inconsistencies that could impede model performance. The pipeline began by selecting only the essential data columns: author, date, repository, and message. The text then underwent normalization, where all messages were converted to lowercase and extraneous whitespace was removed. Following this, irrelevant content, such as automated messages from pull request merges

(e.g., "merge branch") and commits with fewer than three words, was systematically filtered out. The cleaning process was further refined by removing specific patterns that lack semantic value for sentiment analysis, including URLs, email addresses, UUIDs, and metadata tags. Finally, the cleaned messages were tokenized using the DistilBERT tokenizer, and any message exceeding the 512-token input limit of the model was excluded from the dataset.

## 3.3 GOLD STANDARD ANNOTATION AND REFINEMENT

A gold standard dataset of 2,000 commits was established through a rigorous, iterative manual annotation process conducted by two independent annotators. The initial annotation phase utilized a seven-label emotion scheme common in the literature (Joy, Excitement, Satisfaction, Frustration, Anger, Sadness, Neutral). This first pass, however, yielded a Cohen's $\kappa$ score of 0.596, indicating only moderate inter-annotator agreement. A subsequent qualitative analysis of the disagreements revealed that the low score was a result of conceptual ambiguity between closely related emotion categories, such as Joy and Excitement, which proved difficult to distinguish reliably within the technical context of commit messages.

This finding prompted a systematic refinement of the annotation scheme. The seven labels were consolidated into four more robust and distinct categories designed to reduce ambiguity and better capture the developer experience. The revised scheme consisted of **Satisfaction** (merging Joy, Excitement, and gratification), **Frustration** (merging Anger, Sadness, and frustration), **Neutral**, and a newly introduced category, **Caution**. The "Caution" label was specifically created to capture defensive or risk-oriented signals (e.g., warnings, workarounds) that are highly relevant in software development but not captured by traditional emotion models. A second round of annotation using this 4-label scheme resulted in a Cohen's $\kappa$ of 0.710, a score that signifies substantial agreement and confirms the reliability of the refined framework. This iterative process of evaluation and consolidation was critical for producing a high-quality, trustworthy gold dataset for our research.

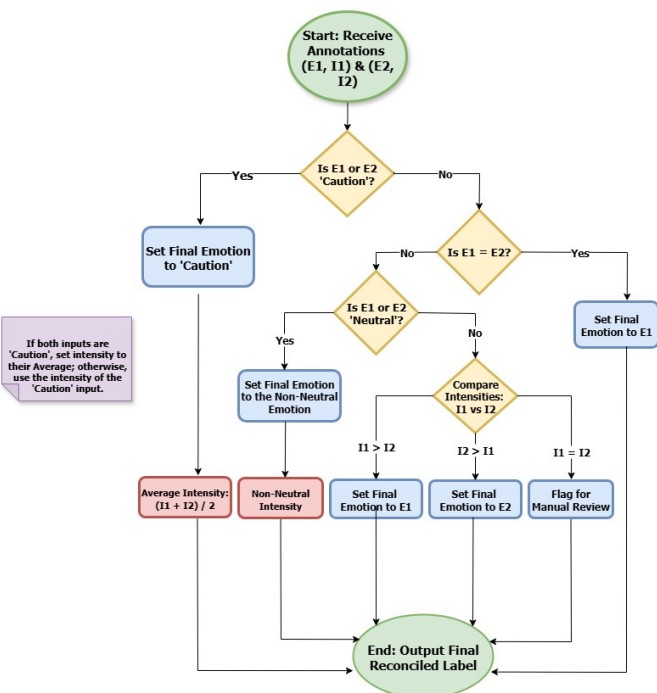

Figure 2: Reconciliation workflow that consolidates two annotations into a final gold label.

# 4 EXPERIMENTAL SETUP AND MODELS

All experiments are conducted on the 2,000-message gold-standard dataset, partitioned into 1,600 training samples and 400 held-out test samples. Performance across all models is reported using Macro F1, which balances class-level contributions and accounts for label imbalance.

## 4.1 BASELINE MODELS AND RATIONALE

To establish a comprehensive set of baselines, we evaluate five pretrained models spanning general-purpose reasoning, code-aware representation, and emotion-focused classification. The lineup includes:

- **RoBERTa-large-MNLI**: a natural language inference model used for zero-shot classification by hypothesis testing.
- **CodeBERT (microsoft/codebert-base)**: pretrained jointly on source code and natural language, representing a domain-specific specialist.
- **DistilRoBERTa Emotion**: trained on multiple emotion datasets, covering fine-grained affect categories.
- **RoBERTa-base GoEmotions**: fine-tuned on the 58k-example GoEmotions dataset, with broad emotion coverage.
- **EmoLLaMA-7B**: a generative LLaMA-based model instruction-tuned for affective tasks.

We exclude `distilbert-base-uncased-emotion` because it lacks a Neutral label, making it incompatible with our taxonomy.

## 4.2 ZERO-SHOT EVALUATION METHODOLOGY

All five baselines are first evaluated in a zero-shot setting. For RoBERTa-large-MNLI, commit messages are treated as premises and paired with hypotheses of the form "This commit expresses {label}," selecting the label with the highest entailment score. For emotion-tuned classifiers, fine-grained outputs (e.g., joy, anger) are mapped into our four-label scheme, with categories such as joy and love mapped to **Satisfaction** and anger or sadness mapped to **Frustration**. The generative model (EmoLLaMA) is prompted with an explicit classification instruction, and its outputs are normalized into the four classes. Finally, a lexicon-based override is applied: if a commit contains predefined caution-related keywords (e.g., "workaround," "warning"), the label is set to **Caution** regardless of model output. This standardized procedure enables fair comparison across heterogeneous architectures.

## 4.3 FINE-TUNING SETUP FOR BASELINE MODELS

Following the zero-shot evaluation, we fine-tune the two strongest candidates: RoBERTa-large-MNLI (generalist) and CodeBERT (specialist), to assess their adaptability to technical text. Both models are trained on the 1,600-sample training set using Hugging Face's `Trainer` API. The classification head is replaced with a four-class output layer, with mismatched dimensions resolved automatically. Training proceeds for five epochs, with checkpoint selection based on validation Macro F1, while external logging is disabled and only the best model retained.

This comparison revealed distinct dynamics: RoBERTa-large-MNLI overfit rapidly, showing limited generalization, whereas CodeBERT maintained stable learning and achieved stronger balanced performance. Accordingly, CodeBERT is adopted as the primary fine-tuned baseline and serves as the reference point for subsequent augmentation experiments in 4.4.

## 4.4 THE COMMITUNE METHOD

We propose **CommiTune**, a hybrid augmentation pipeline that combines generative large language models with code-aware fine-tuning to overcome the scarcity of labeled developer emotion data. The method consists of three steps.

### 4.4.1 Fine-Tuning the Generative Model

We adapt `Meta-LLaMA-3.1-8B-Instruct` on the 1,600 manually labeled training commits to align the generator with the four-label taxonomy. Fine-tuning is performed with Hugging Face's PEFT and LoRA adapters to ensure efficiency. The following hyperparameters are used: learning rate $= 2 \times 10^{-5}$, batch size = 16, AdamW optimizer with weight decay 0.01, linear learning rate scheduler, and 3 training epochs with early stopping based on validation loss. The model's classification head is resized for four outputs, consistent with our taxonomy.

### 4.4.2 Generative Data Augmentation

Using the fine-tuned LLaMA, we generate two paraphrases for each training commit, yielding an augmented dataset of ∼4,800 examples. Prompts are structured as follows:

```
You are tasked with paraphrasing commit messages while preserving their emotional label.
Label:   <LABEL>
Commit:  <ORIGINAL_COMMIT>
Output two alternative paraphrases of the commit message
that preserve the same meaning and emotional labels.
```

Generated outputs are post-processed to remove duplicates, normalize whitespace, and enforce single-sentence formatting. This augmentation increases lexical and syntactic diversity while maintaining the integrity of the gold-standard labels.

### 4.4.3 Final Model Training

The augmented dataset is used to retrain CodeBERT, selected as the strongest baseline in 4.3. The 4,800 examples are partitioned using an 80–20 split, with the 20% subset serving as a held-out validation set. Training mirrors the setup in 4.3: five epochs, cross-entropy loss, early selection by Macro F1, logging disabled, and only the best checkpoint retained.

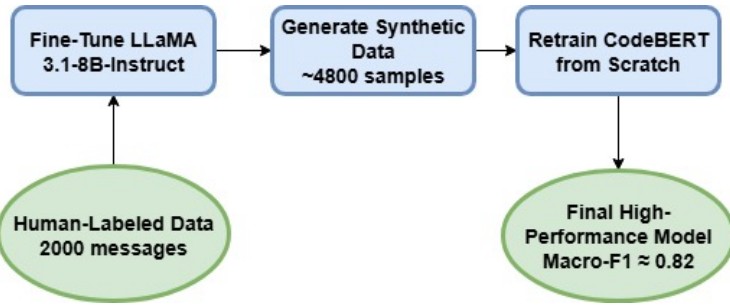

Figure 3: Commitune Method Diagram.

This pipeline integrates the strengths of generative models (diverse paraphrasing) and domain-specific encoders (stable code-aware representations). CommiTune thereby offers a reproducible and scalable method for technical emotion detection.

## 5 Results

This section presents the empirical evaluation of our proposed method. We first establish a set of strong baselines by assessing the performance of both zero-shot (4.2) and fine-tuned approaches (4.3). We then report the results of our CommiTune pipeline, demonstrating its effectiveness in improving developer emotion classification (4.4).

### 5.1 Baseline Performance

The results in Table 1 clearly show that off-the-shelf zero-shot models operate at near-chance performance when applied to commit message sentiment analysis. With four target categories, a

Table 1: Baseline zero-shot model performance on commit sentiment analysis.

| Model | Macro-F1 |
|---|---|
| RoBERTa-large-MNLI | 0.2058 |
| CodeBERT | 0.1968 |
| DistilRoBERTa | 0.1886 |
| RoBERTa-base GoEmotions | 0.1374 |
| EmoLLaMA-7B | 0.1296 |

random classifier would be expected to achieve a Macro-F1 of 0.25; the best baseline, RoBERTa-large-MNLI, reaches only 0.2058. CodeBERT shows some sensitivity to software-specific phrasing but still fails to capture affective nuance, while emotion-tuned classifiers such as DistilRoBERTa and GoEmotions underperform due to domain mismatch. The generative model EmoLLaMA-7B produces fluent categorical outputs, yet its predictions are poorly calibrated and unstable. These results reinforce the need for domain-adapted approaches including fine-tuning, label consolidation, and data augmentation to reliably identify subtle emotions like Satisfaction, Frustration, and Caution in highly technical commit text.

Table 2: Fine-tuned model performance across emotion categories.

| Class | Precision | Recall | F1-Score | Support |
|---|---|---|---|---|
| Caution | 0.79 | 0.79 | 0.79 | 53 |
| Satisfaction | 0.47 | 0.44 | 0.46 | 107 |
| Neutral | 0.58 | 0.63 | 0.60 | 139 |
| Frustration | 0.52 | 0.50 | 0.51 | 101 |
| Macro Avg | 0.59 | 0.59 | 0.59 | 400 |
| Accuracy | – | – | 0.56 | 400 |

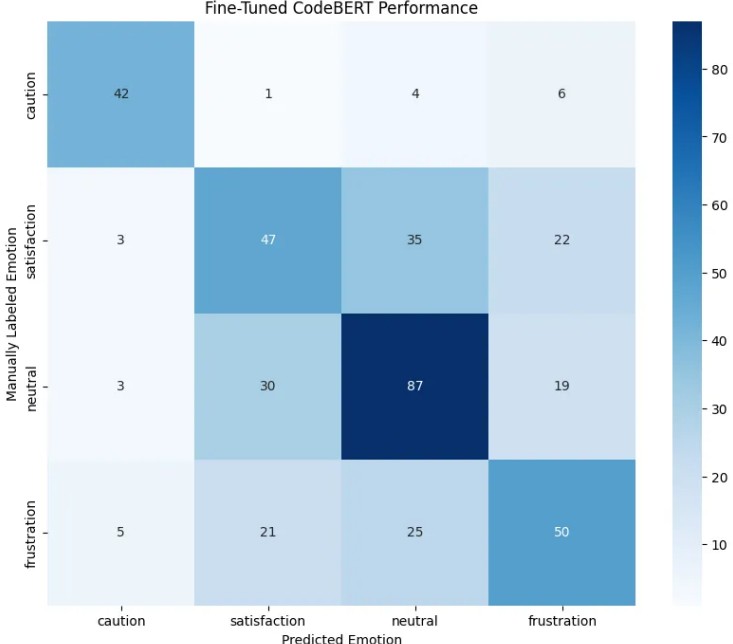

Figure 4: Confusion matrix for the fine-tuned CodeBERT baseline.

The results in Table 2 and Figure 4 show that off-the-shelf zero-shot models operate at near-chance performance on commit message sentiment analysis. With four target categories, a random classifier

would achieve a Macro-F1 of 0.25; the best baseline, RoBERTa-large-MNLI, reaches only 0.2058. CodeBERT shows some sensitivity to software-specific phrasing but still fails to capture affective nuance, while emotion-tuned classifiers such as DistilRoBERTa and GoEmotions underperform due to domain mismatch. These results reinforce the need for domain-adapted approaches including fine-tuning, label consolidation, and data augmentation.

## 5.2 COMMITUNE PERFORMANCE

To evaluate our proposed CommiTune pipeline, CodeBERT was retrained on the augmented dataset and performance was assessed on the same held-out test set of 400 manually-labeled samples.

Table 3: CodeBERT performance after retraining with the CommiTune pipeline.

| Emotion | Precision | Recall | F1-Score |
|---|---|---|---|
| Frustration | 0.87 | 0.79 | 0.83 |
| Satisfaction | 0.74 | 0.81 | 0.77 |
| Neutral | 0.80 | 0.80 | 0.80 |
| Caution | 0.90 | 0.88 | 0.89 |

Retraining CodeBERT on the augmented dataset resulted in a dramatic improvement across all metrics. As shown in Table 3, the CommiTune approach achieves an overall accuracy of 81%, a 25-point increase from the baseline, and a Macro F1-score of 0.82, a 23-point increase over the 0.59 baseline.

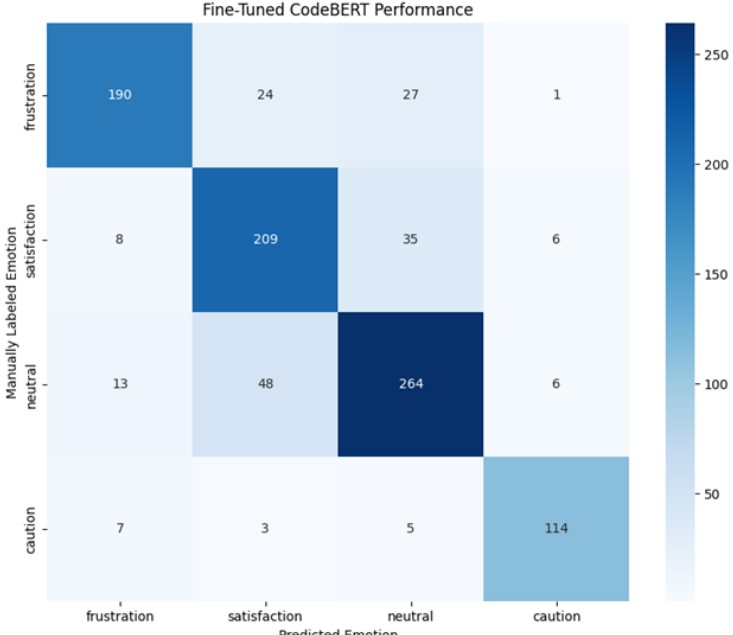

Figure 5: Confusion matrix for CommiTune (CodeBERT trained with augmented data).

Crucially, the model's primary weakness in identifying the Satisfaction class has been resolved. The F1-score for Satisfaction jumped from a poor 0.46 to a strong 0.77. The confusion matrix in Figure 5 reveals that the model no longer systematically misclassifies Satisfaction commits as Neutral, demonstrating a more nuanced understanding of positive affective signals. Furthermore, the model exhibits a significant reduction in off-diagonal errors across all categories. The performance on Frustration (F1: 0.83) and Caution (F1: 0.89) is particularly strong, indicating a more robust and reliable classifier overall. This demonstrates that the generative data augmentation strategy was the key to repairing the representational gaps in the baseline model, unlocking state-of-the-art performance on this nuanced, domain-specific task.

## 6 CONCLUSION

In this work, we addressed the challenge of detecting developer emotions in technical commit messages, a domain where general-purpose language models, trained on a mix of conversational text and informal code repositories, systematically fail.

We introduced a rigorously annotated dataset for software engineering affect and demonstrated that while a fine-tuned CodeBERT provides a strong baseline, its performance is significantly constrained by data scarcity. Our proposed hybrid pipeline, **CommiTune**, overcomes this limitation by using a fine-tuned LLaMA model for data augmentation, achieving a state-of-the-art Macro F1-score of 0.82 and substantially outperforming all baselines, particularly in resolving the ambiguities of the Satisfaction class.

This result offers a broader lesson for specialized NLP: hybrid augmentation presents a practical and effective strategy to bridge the representational gaps in domains where labeled data is scarce. While our study focuses on English-language commits from professional repositories, future work could explore this method's efficacy in multilingual software projects or adapt it to other technical fields like legal or biomedical text analysis.

By releasing our dataset, code, and models, we provide a reproducible benchmark to spur further research into domain-adapted affective modeling—a feasible and socially impactful field with clear applications in improving developer well-being and software reliability.

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
