# OpenReview forum: "Improving Developer Emotion Classification via LLM-Based Augmentation"
_ICLR.cc/2026/Conference — Submitted to ICLR 2026_

### Official Review · Reviewer_tzMf · 2025-10-30

**Soundness:** 2
**Presentation:** 2
**Contribution:** 2
**Rating:** 4
**Confidence:** 4

**Summary:**

The paper builds a 2,000-commit gold set with a domain-specific 4-label scheme (Satisfaction, Frustration, Caution, Neutral), reports near-chance zero-shot performance for several pretrained models, then proposes CommiTune: fine-tune LLaMA to paraphrase label-preserving commits and re-train CodeBERT on the augmented set, improving Macro-F1 from ~0.59 to ~0.82 on a 400-sample test split. While the task is meaningful, the methodology largely follows a standard supervised pipeline; key evaluation choices (baseline fairness, augmentation leakage, statistics) limit the strength of claims.

**Strengths:**

1.Addresses an under-explored technical domain; defines a domain-tailored 4-label scheme including Caution.

2.Clear end-to-end pipeline with reproducible ingredients (dataset, prompts, mappings).

3.Empirically shows large gains of hybrid augmentation over a code-aware baseline (Macro-F1 ~0.82).

**Weaknesses:**

1.Zero-shot protocol confound: a lexicon override assigns Caution regardless of model output—compromising baseline fairness. Report with/without override.

2.Augmentation leakage risk: label-preserving paraphrases without similarity filtering can inflate scores; no audit vs test set is reported.

3.Underspecified methods: LLaMA “classification head resized” is atypical for a generative setup—needs architectural/IO details.

4.Insufficient baselines/ablations: no comparison to other text augmentation (e.g., back-translation/EDA), no sensitivity to #paraphrases, and no multi-seed CIs.

5.Sampling rationale unclear: path from millions → thousands lacks principled selection criteria (coverage, class balance, temporal/project split).

**Questions:**

1.Provide zero-shot results with vs without the lexicon override; quantify its class-wise impact.

2.Report a leakage audit: n-gram/embedding similarity between augmented train and test; show filtered near-duplicates.

3.Clarify LLaMA usage: is it trained as a generator only, or also as a classifier (details of the “resized head”)?

4.Add ablations: number of paraphrases, alternative augmentation (back-translation), and augmentation-free vs augmentation-with-filtering.

5.Provide statistical robustness (3–5 seeds, mean±std, bootstrap/McNemar tests).

6.Justify the sampling funnel (coverage/imbalance control, cross-project/temporal splits) from 4.3M to 2k; discuss generalization.

---

### Official Review · Reviewer_tKF6 · 2025-10-31

**Soundness:** 3
**Presentation:** 2
**Contribution:** 3
**Rating:** 4
**Confidence:** 3

**Summary:**

This paper looks at a unique and tricky problem of finding emotions in GitHub commit messages. Normal emotion labels like joy or anger don't really fit the professional tone of developers. So the authors make a new dataset of 2,000 human-labeled commits, using four custom classes: Satisfaction, Frustration, Caution, and Neutral. They test big language models but find they perform poorly because these models don't understand the coding domain or the special emotion labels. To fix this, they build a system called CommiTune, which fine-tunes a llama model on the labeled commits and uses that model to make two paraphrased versions of each commit. The authors then train CodeBERT again on this bigger dataset. This process improves the F1 score.

**Strengths:**

- I like the choice of problem of detecting emotions in developer commit messages. It can help with things like spotting burnout, predicting bugs, and keeping teams healthy. Yet, normal emotion models don’t handle this area yet.
- The data is big, and the collection process is clear and transparent making it easy to reproduce in the future.
- CommiTune is simple but interesting. It tries to combine the best of both worlds of LLM creativity and code model's accuracy by rephrasing the messages while keeping same labels.
- The results improve when author's proposed method is incorporated, marking the work as influential.

**Weaknesses:**

- The LLM-made paraphrases may sound different but still repeat the same wording patterns from the original data. For example, if “fixed X” means Satisfaction, the model might just keep generating versions of that line, creating repetition instead of real variety.
- All commits are from big GitHub projects like Python or TensorFlow. These are well-written, professional messages. The method might not work well for smaller projects, or commits with typos, slang, or mixed languages.
- The baseline models tested are mostly general emotion detectors or CodeBERT. They didn’t include software-specific emotion models like SentiMoji [1] or EmoD [2].
- The Caution label may be a bit overused. The paper sometimes forces words like "warning" to mean Caution, even when in context it could be Neutral, for e.g. in normal system logs. This might make the results slightly biased.

[1] Zhenpeng Chen, Yanbin Cao, Xuan Lu, Qiaozhu Mei, and Xuanzhe Liu. 2019. SEntiMoji: an emoji-powered learning approach for sentiment analysis in software engineering. In Proceedings of the 2019 27th ACM Joint Meeting on European Software Engineering Conference and Symposium on the Foundations of Software Engineering (ESEC/FSE 2019). Association for Computing Machinery, New York, NY, USA, 841–852. https://doi.org/10.1145/3338906.3338977
[2] K. P. Neupane, K. Cheung and Y. Wang, "EmoD: An End-to-End Approach for Investigating Emotion Dynamics in Software Development," 2019 IEEE International Conference on Software Maintenance and Evolution (ICSME), Cleveland, OH, USA, 2019, pp. 252-256, doi: 10.1109/ICSME.2019.00038. keywords: {Tools;Task analysis;Databases;Data collection;Software;Time series analysis;Software engineering;Emotion awareness, emotion dynamics, emotion intensity, software project team, time series database},

**Questions:**

- Did you check if the LLM’s paraphrases changed the meaning by mistake? For example, "refactored module for clarity" (a happy tone) could turn into "rewrote messy code" (a frustrated tone).
- How much does CommiTune depend on which LLM you use? Would a smaller model like LLaMA-3-8B or Mistral-7B give similar improvements?
- Can the Caution label be split into smaller parts which may need different handling in real projects.? Like one for "temporary fix or workaround" and another for "serious security issue"?
- Did you think about multi-task learning, where the model predicts both emotion and commit type (bug fix, feature, refactor)? That might make it more stable and accurate.
- Is Satisfaction really an emotion, or just a sign that the job is done? If it's mostly about completion, maybe simple clues like the word "implemented" could already tell that.

---

### Official Review · Reviewer_KLi7 · 2025-11-01

**Soundness:** 3
**Presentation:** 3
**Contribution:** 3
**Rating:** 4
**Confidence:** 3

**Summary:**

This paper examines emotional classification across more than 2,000 GitHub "commit messages that have been human-labeled with a four-label scheme". The core method, CommiTune, fine-tunes a LLaMA model to generate paraphrased training data and then fine-tunes CodeBERT on the expanded set.  Zero-shot baselines are near-chance (Macro-F1 ≈ 0.13–0.21), a fine-tuned CodeBERT baseline reaches Macro-F1 ≈ 0.59, and CommiTune lifts performance to Macro-F1 ≈ 0.82 (Accuracy ≈ 0.81), with big gains especially on the Satisfaction class.

**Strengths:**

This paper has merit in these points:
1. Clear, domain-motivated problem. This work focuses on developers' emotional analysis from GitHub commits and proposes a compact 4-label scheme (Satisfaction, Frustration, Caution, Neutral) tailored to software engineering discourse.

2. Clear design of pipeline: This work starts from a 4.3M-commit Kaggle corpus, the authors use stratified sampling (by repo, project size, and quarter) to form a 20k subset, then produce a 2k gold set with a documented preprocessing pipeline that removes automated/low-information messages. This pipeline is clearly articulated and appropriate for the task.

3. Simple, reproducible hybrid augmentation. The CommiTune pipeline—fine-tune an LLaMA variant on labels → generate paraphrases → fine-tune CodeBERT—balances practicality and impact. The resulting jump from a CodeBERT baseline Macro-F1 0.59 (Acc 0.56) to Macro-F1 0.82 (Acc 0.81) on the same held-out test split is large and meaningful, especially for the previously weak Satisfaction class.

**Weaknesses:**

This paper does have limitations and weaknesses in the following parts:
1. The text states “two paraphrases per training commit, yielding ~4,800 examples,” which suggests 1,600 originals + 3,200 paraphrases. Please disambiguate whether the original 1,600 are included and how class balance is controlled after augmentation.

2.  Results appear to be single-run point estimates. Confidence intervals or multiple seeds (with mean±std) and significance tests  ( like bootstrap on Macro-F1 or accuracy) are needed, given the relatively small test set (n=400).

3. Minor methodological inconsistencies. The LLaMA fine-tuning description mentions resizing a "classification head" despite ultimately using the model for **generation/paraphrase**. This could be clarified or corrected to prevent confusion.

**Questions:**

Here are questions to the authors for rebuttal

1. Was the 1.6k/400 split repo-disjoint and/or time-disjoint? If not, can you report results with (a) repo-level disjoint splits, and (b) a chronological split (train on earlier quarters, test on later)? This would directly address topical and temporal leakage concerns.

2. I also would like to clarify this: Do the 4,800 training examples include the original 1,600 commits or only the 3,200 paraphrases? What is the exact class distribution before/after augmentation, and did you cap synthetic samples per class to prevent imbalance or drift?

3. Please report zero-shot results without the caution-lexicon override, and/or present the override as a separate lexicon baseline.

4. I'm sceptical of the model-specific performance. Could you include at least one modern code/text encoder (e.g., CodeT5+, UniXcoder) and a strong general text encoder baseline to reinforce that the gains are not CodeBERT-specific?

---

### Official Review · Reviewer_tdxY · 2025-11-04

**Soundness:** 2
**Presentation:** 2
**Contribution:** 2
**Rating:** 2
**Confidence:** 4

**Summary:**

The paper tackles developer emotion classification in technical commit messages, arguing that general-purpose langauge models and emotion taxonomies fail in this specialized context. To address this, the authors make two main contributions. First, they introduce a dataset of 2K GitHub commit messages manually annotated with a novel, domain-specific four-label scheme. Second, the paper proposes CommiTune, a three step method composed of (1) fine-tuning a LLaMA-3.1-3B model on the training set (2) use the model to generate emotion-aware paraphrased synthetic examples for training commits and (3) Training a CodeBERT from scratch on this newly augmented dataset. The paper claims that this pipeline achieves a state-of-the-art Macro-F1 score of 0.82.

**Strengths:**

- The creation of the 2,000-sample labeled dataset is a solid contribution to the software engineering NLP community. The authors offer great insights into their annotation process, particularly how they iterated on the label scheme to improve their Cohen's $\kappa$ score.

**Weaknesses:**

While the paper is well-motivated on paper and the 23-point F1 jump looks impressive on the surface, a deeper look reveals methodological flaws, missing baselines, and a dataset that is too small and biased to support the paper's broad claims.

- The paper claims in Section 4.4.1 that it fine-tunes the LLaMA model as a classifier, explicitly stating it "resized [the] classification head for four outputs". But then, in the very next step, it says it uses this same "fine-tuned LLaMA" for a generative paraphrasing task (Section 4.4.2). This is a massive contradiction. How can a model trained with a classification head be used for generation? The LoRA adapters were trained to classify (i.e., collapse text into one of four labels), so they would almost certainly damage the model's creative paraphrasing ability, not help it. The paper never explains this or justifies why it chose this convoluted pipeline over a much simpler and more logical few-shot prompting baseline.
- **The paper is missing crucial baselines and ablations to prove its method actually works**. (1) Why not just use the fine-tuned LLaMA model directly as the classifier? The paper never reports this, so we have no idea if their complex 3-step pipeline is any better than a single-step one. The only logical reason to use this pipeline (a big LLaMA to teach a "lightweight" CodeBERT ) would be for inference efficiency, but the paper never mentions efficiency, model size, or deployment costs. (2)  The 23-point F1 gain is attributed to augmentation, but they fail to justify key hyperparameters. Why generate "two paraphrases"? What happens with one, or five, or ten?  (3) Fine-tuning a 8B model on only 1,600 samples is also a red flag for memorization. The generator was likely just creating stereotyped, low-variance copies of the training data, which made the task artificially easier for CodeBERT rather than actually solving the "representation gap". Analysis into how these rephrased examples look like are missing.

- **Dataset issues** (1) The dataset is small (2,000 samples) and heavily biased. The authors only sampled from 34 "popular" projects and explicitly filtered out "casual or hobbyist" code. This would be okay if the evaluation setup would include a broader distribution. (2) The "Caution" label isn't an emotion. The paper defines it as capturing "defensive or risk-oriented signals (e.g., warnings, workarounds)". This is a semantic category, not an affective state. (3) The annotation workflow is biased. The reconciliation process (Figure 2) hard-coded a rule: if either annotator said "Caution," that label would win the disagreement. This explains the suspiciously high 0.79 F1 baseline for that class —it's an artifact of a biased labeling process, not a learned feature.

- This "Caution" artifact bleeds into the evaluation. The authors admit to using a "lexicon-based override" for "Caution" (e.g., "workaround," "warning") in their zero-shot evaluation. It is unclear if this keyword-matching rule was used in the final baseline and CommiTune experiments. Given the high F1 score, it's very likely it was, which means a top-scoring part of the model is just learning a simple pattern. At the same time, the paper does not cite the lexicon used or describe the lexicon if it was created by the authors.

- The paper is framed as a way to "address developer burnout" , but there are no analyses in the paper that actually connect the 'Frustration' label to any real-world measure of burnout.

- Overall, the novelty is very limited and to me the paper does not dig deep enough into the problem through an OOD evaluation or through investigating how the proposed approach could  tackle “addressing developer burnout” (e.g., apply your method in a large-scale experiment or carry out additional annotations to quantify this)? As it stands, this dataset introduction is not well-enough motivated.

**Questions:**

See above.

---

### Public Comment · ~Wenqi_Marshall_Guo1 · 2025-11-13
**Is there ethical concerns for infer developer's emotion by commit messages?**

I would like to raise an ethical consideration regarding the task of inferring developer emotion from commit messages. Commit messages are primarily written for technical coordination rather than personal disclosure, and using them as signals of individual emotional state introduces privacy risks. It also creates potential for discriminatory outcomes if such inferences are used in evaluation or management contexts, especially given the systematic misclassification patterns described in this paper.

---

### Meta-Review · Area_Chair_hUm5 · 2025-12-28

**Summary:**

The paper tackles developer emotion classification in technical commit messages, arguing that general-purpose langauge models and emotion taxonomies fail in this specialized context. They introduce a dataset of 2K GitHub commit messages manually annotated with a novel, domain-specific four-label scheme, and then proposes a pipeline of CommiTune for classification. The paper claims that this pipeline achieves a state-of-the-art Macro-F1 score of 0.82.

**Reviewer Concerns:**

Advantages:
1. The task is new and the authors address an under-explored technical domain; this work defines a domain-tailored 4-label scheme including Caution.

2.Clear end-to-end pipeline with reproducible ingredients (dataset, prompts, mappings). The dataset is big (2000 samples), and the collection process is clear and transparent.

3. Simple, reproducible hybrid augmentation. The CommiTune pipeline—fine-tune an LLaMA variant on labels → generate paraphrases → fine-tune CodeBERT—balances practicality and impact. Empirically shows large gains of hybrid augmentation over a code-aware baseline (Macro-F1 ~0.82).

Disadvantages:

1. Methodological flaws: reviewers argue there are some flaws in using different discriminative or generative models.
2. Missing baselines: The paper is missing crucial baselines and ablations to prove its method actually works.
3. Evaluation： the dataset is too small and biased to support the paper's broad claims. The annotation is also biased. Multiple runs are necessary for reporting the results.

**Reviewer Scores:**

the ratings come out as 2,4,4. no author responses were provided.

---

### Decision · Program_Chairs · 2026-01-26

Reject